# Matrix Remodeling and Hyaluronan Production by Myofibroblasts and Cancer-Associated Fibroblasts in 3D Collagen Matrices

**DOI:** 10.3390/gels6040033

**Published:** 2020-09-30

**Authors:** Jiranuwat Sapudom, Claudia Damaris Müller, Khiet-Tam Nguyen, Steve Martin, Ulf Anderegg, Tilo Pompe

**Affiliations:** 1Laboratory for Immuno Bioengineering Research and Applications, Division of Engineering, New York University Abu Dhabi, Abu Dhabi P.O. Box 129188, UAE; 2Institute of Biochemistry, Faculty of Life Sciences, Universität Leipzig, 04103 Leipzig, Germany; damaris.mueller@tu-dresden.de (C.D.M.); steve.martin@uni-leipzig.de (S.M.); tilo.pompe@uni-leipzig.de (T.P.); 3Department of Dermatology, Venereology and Allergology, Faculty of Medicine, Universität Leipzig, 04103 Leipzig, Germany; tam.nguyen@medizin.uni-leipzig.de (K.-T.N.); ulf.anderegg@medizin.uni-leipzig.de (U.A.)

**Keywords:** hyaluronan, tumor microenvironment, transforming growth factor beta 1, matrix remodeling, myofibroblast, cancer associated fibroblast

## Abstract

The tumor microenvironment is a key modulator in cancer progression and has become a novel target in cancer therapy. An increase in hyaluronan (HA) accumulation and metabolism can be found in advancing tumor progression and are often associated with aggressive malignancy, drug resistance and poor prognosis. Wound-healing related myofibroblasts or activated cancer-associated fibroblasts (CAF) are assumed to be the major sources of HA. Both cell types are capable to synthesize new matrix components as well as reorganize the extracellular matrix. However, to which extent myofibroblasts and CAF perform these actions are still unclear. In this work, we investigated the matrix remodeling and HA production potential in normal human dermal fibroblasts (NHFB) and CAF in the absence and presence of transforming growth factor beta -1 (TGF-β1), with TGF-β1 being a major factor of regulating fibroblast differentiation. Three-dimensional (3D) collagen matrix was utilized to mimic the extracellular matrix of the tumor microenvironment. We found that CAF appeared to response insensitively towards TGF-β1 in terms of cell proliferation and matrix remodeling when compared to NHFB. In regards of HA production, we found that both cell types were capable to produce matrix bound HA, rather than a soluble counterpart, in response to TGF-β1. However, activated CAF demonstrated higher HA production when compared to myofibroblasts. The average molecular weight of produced HA was found in the range of 480 kDa for both cells. By analyzing gene expression of HA metabolizing enzymes, namely hyaluronan synthase (HAS1-3) and hyaluronidase (HYAL1-3) isoforms, we found expression of specific isoforms in dependence of TGF-β1 present in both cells. In addition, *HAS2* and *HYAL1* are highly expressed in CAF, which might contribute to a higher production and degradation of HA in CAF matrix. Overall, our results suggested a distinct behavior of NHFB and CAF in 3D collagen matrices in the presence of TGF-β1 in terms of matrix remodeling and HA production pointing to a specific impact on tumor modulation.

## 1. Introduction

Cancer-associated fibroblasts (CAF) are one of the key players in regulating cancer progression and metastasis [1,2]. They are known as activated fibroblasts or myofibroblasts and are characterized by an extensive expression of alpha-smooth muscle actin (αSMA) and high secretion of a broad spectrum of cytokines and extracellular matrix (ECM) components, which promote the growth and invasion of tumor cells [2,3]. The difference between CAFs and myofibroblasts is often considered functional, rather than biological markers [4]. Current reports demonstrated that CAF also produce classical wound-healing mediators [5]. However, there is evidence showing epigenetic and genetic distinction between CAF and wound-healing related myofibroblasts which are present during wound repair calling for a more differentiated view on phenotype and function of CAF and myofibroblasts [6]. The fibroblast differentiation is tightly regulated by transforming growth factor-beta 1 (TGF-β1) which is a prominent cytokine found in the tumor microenvironment and during tissue repair processes [7,8]. In wound healing, TGF-β1-activated fibroblasts differentiate towards myofibroblasts which can produce new ECM components and reorganize the matrix microarchitecture [9,10]. In the tumor microenvironment, TGF-β1 may suppress tumor growth during the pre-malignant stage, while it promotes matrix remodeling and matrix stiffening by myofibroblasts which can assist initiation of cancer cell invasion and proliferation [11,12].

One of the ECM components that has been discussed to be an important player in cancer progression is hyaluronan (HA) [13,14]. HA is a non-sulfated glycosaminoglycan composed of D-glucuronic and N-acetyl-D-glucosamine. Especially in solid tumors, higher accumulation of HA can be observed which correlated to aggressive malignancy both in preclinical models and in patients [15,16]. HA molecules occupy a large hydrodynamic volume and can associate with collagen and proteoglycans in the ECM. The turnover of HA in the tissue is 1 to 2 days in the epidermis [17] and it is tightly regulated by two key enzyme classes; hyaluronan synthases (HAS) and hyaluronidases (HYAL). The synthesis of HA is accomplished by three HAS isoforms, namely HAS1, HAS2 and HAS3, which produce different sizes of HA. At physiological conditions, HAS1 and HAS2 produce HA with average molecular weights of about 200 kDa to 2000 kDa, while HAS3 yielded HA product at the average molecular weight of 100 to 1000 kDa [18,19]. HAS2 is the major isoform responsible for the HA synthesis in fibroblastic cells and its deficiency leads to the embryonic lethality [20]. In addition, an overexpression of HAS2 caused an enhanced migration of fibroblasts and severe fibrosis in mouse model [21,22,23]. Besides HA synthesis, HA degradation plays an important role in the HA-mediated cell response in wound healing and cancer progression. HA degradation is regulated either by free radical-related depolymerization occurring in the presence of reactive oxygen species (ROS), Maillard products, enzymatically by hyaluronidases (HYAL1-3) or other enzymes e.g., PH20, CEMIP, TMEM2. In humans three isoform of hyaluronidases, namely HYAL1, HYAL2 and HYAL3, were reported. However, HYAL-1 and HYAL-2 constitute the major hyaluronidases of somatic tissues. HYAL1 has been reported to play an important role in the hydrolysis of HA, resulting in a small HA-oligosaccharides which demonstrated to promote proliferation and invasion of cancer cells. Many effects of HA in tumor biology depend on HA size and concentration, as well as presented form, and they rely on the interaction with its main cellular receptors CD44 (and tumor specific splice variants thereof) and HA-mediated motility receptor (RHAMM) of the cancer cells [24,25,26].

Current research on cancer behavior is shifting towards more physiologically and pathologically relevant three-dimensional (3D) cell culture models, rather than the conventional two-dimensional planar tissue culture plastic [27,28,29]. 3D fibrillar collagen matrices are a versatile platform to adjust composition, microstructure and mechanics of 3D cell culture scaffolds and are widely used to mimic interstitial tissue [28,30]. Also, there have been works on how HA affects cancer cell behavior in the cancer microenvironment using 3D collagen matrices [14,31]. However, less is known about the differential behavior of wound-healing related myofibroblasts and CAF in term of proliferation, matrix remodeling potential, and HA production in 3D cell culture settings. In order to address both, normal human dermal fibroblasts (NHFB), and CAF were embedded into well-defined 3D collagen matrices mimicking the ECM of a tumor microenvironment and were activated thereafter by TGF-β1. Cell proliferation, matrix remodeling and HA production were studied and compared between both cell types.

## 2. Results and Discussion

Wound-healing related myofibroblasts and activated cancer-associated fibroblasts (CAF) appear to have common phenotypical features in regulating matrix remodeling and HA accumulation. However, the discrepancy of both cells is still unclear. In this work, we aimed to investigate and compare the capability of HA production and of normal human fibroblast (NHFB) and CAF from human skin of malignant melanoma. For CAF, skin adjacent to the tumor (<5 mm) was used, whereas NHFB were prepared from female breast skin. The use of the CAF from the adjacent tumor microenvironment is due to the knowledge that these cells can affect tumor progression by preparing specific microenvironment for cancer cell to invade and proliferate [32]. Especially in the malignant melanoma of human skin, it has been demonstrated that tumor cells induce HA synthesis in peritumoral fibroblasts and in turn regulate melanoma proliferation [33]. For the experimental setup, both cells were embedded in 3D collagen matrices reconstituted at concentration of 2 mg/mL collagen I and were cultured for seven days in the presence and absence of 10 ng/mL TGF-β1. Afterwards, cells were analyzed in term of proliferative capacity, alpha smooth muscle actin (aSMA) expression, matrix remodeling and the capacity of HA production. Additionally, HA molecular weight and gene expression of HA metabolism were quantified.

### 2.1. TGF-β1 Enhanced Matrix Stiffening, But Did Not Affect Proliferation and aSMA Expression in CAF

TGF-β1 is known to be a key regulator of fibroblast proliferation and myofibroblast differentiation [10,34]. To demonstrate the effect of TGF-β1 on NHFB and CAF, both cells were cultured in the presence and absence of 10 ng/mL TGF-β1 for seven days. We first analyzed the proliferation of both cells using commercial WST-1 assay. As shown in Figure 1A, cell proliferation of NHFB was enhanced 1.8-fold in response to TGF-β1 when compared to untreated counterparts. An increase in cell proliferation of NHFB in the presence of TGF-β1 concurs with previous reports [11,35]. On the contrary, CAF exhibited a higher proliferation when compared to untreated NHFB but did not show an increase in cell proliferation with TGF-β1 in this regard. This might be explained by a metabolic reprogramming of CAF in prolonged exposure to TGF-β1 in the tumor microenvironment [6,36,37,38]. Another possible explanation could be that CAF secrete a high basal level of TGF-β1 [39,40,41], which might act in a paracrine fashion inducing maximum proliferation level. It has been also demonstrated that the TGF-β1 receptor and its downstream signaling via SMAD2/3 might be altered in CAF [42]. In addition, other soluble factors e.g., fibroblast growth factor [43] and the differential expression level of cell cycle related genes [44] could also affect the proliferative behavior of CAF.

We next studied the *aSMA* expression as a prominent marker of activated fibroblastic cells and their impact in matrix remodeling. In addition, aSMA is widely used as a CAF marker, since there is no specific marker that is exclusively expressed by CAFs. As shown in Figure 1B, representative cell morphological appearance of NHFB and CAF cultured with and without TGF-β1 presence is depicted. Actin fibers could be seen in CAF with and without TGF-β1 treatment, while thicker actin fibers could be observed in TGF-β1 treated NHFB (myofibroblasts). This observation is in line with our previous report [11,45] and supports the differentiation of fibroblasts into myofibroblasts. To confirm the myofibroblast phenotypes, we investigated the gene expression of *aSMA* using real-time quantitative polymerase chain reaction (RT-qPCR) (Figure 1C). We found a significantly higher expression of *aSMA* in CAF with and without treatment with TGF-β1 when compared to NHFB and myofibroblasts. However, the gene expression level of *aSMA* did not enhance further in TGF-β1 treated CAF, while TGF-β1 treated NHFB showed a significant increase in this regard. A recent report demonstrated that aSMA expression of pancreatic fibroblast is dependent on collagen fibril content coated on 2D coverslip [46], while fibril density of 3D collagen matrices appears to suppress fibroblast differentiation as demonstrated by a reduction in aSMA expression of NHFB [45]. The enhanced gene expression of aSMA in myofibroblasts is due to the increased number of aSMA expressing cells (Figure 1D,E), similar to our previous report [35,47]. Surprisingly, the number of aSMA positive cells in CAF did not change in response to TGF-β1. This result confirmed that our isolated cells exhibited a steady CAF marker. The expression of aSMA enhances the contractile ability of activated fibroblastic cells during tissue repair, enhancing tissue density and tissue elasticity which in turn contribute to the wound closure [9,35,48,49]. On the contrary, an increase of aSMA expressing cells in cancerous tissue also correlates to the fibrotic-like tissue stiffening but it is related to metastatic occurrence and poor prognosis [50].

To correlate the *aSMA* expression with matrix remodeling, we analyzed mechanical properties and matrix porosity of 3D collagen matrices. The elastic modulus of collagen matrices was measured by colloidal probe force spectroscopy techniques using atomic force microscopy. As show in Figure 2A, both NHFB and CAF matrices exhibited similar matrix stiffness. TGF-β1 stimulated a cell-mediated matrix stiffness increase for both cell types. However, TGF-β1-treated CAF matrices appear to have slightly higher stiffness than their NHFB counterparts. Since the mechanical properties of the matrices are coupled to the topological parameters, we further analyzed pore size of decellularized matrices using custom-built image analysis toolbox (Figure 2B). Collagen matrices reconstituted at concentration of 2 mg/mL exhibited a mean pore size at 8.6 ± 0.6 µm. Culturing both cells on collagen matrices reduced the pore size of the collagen matrices (Figure 2B and see also Figure 3C). In the matrices of TGF-β1 treated cells, the porosity reduced when compared to the untreated counterparts. Interestingly, CAF matrices demonstrated larger mean pore size when compared to NHFB matrices, while their matrix stiffness was at the similar level as the NHFB matrices. This result suggests that the stiffening in CAF matrix might arise from a change in matrix composition or crosslinking, but not because of a decrease in pore size [28]. We therefore analyzed the expression of alpha chain of the type 1 collagen (*Coll1a1*) and fibronectin with EDA-containing domain (*EDA-FN*). As shown in Figure 2C,D, in the presence of TGF-β1 only myofibroblast enhanced the expression of *Coll1a1* and *EDA-FN*, while the expression level of both matrix protein remains xsimilar in CAF with and without TGF-β1 presence. However, the level of *Coll1a1* and *EDA-FN* expression in CAF is higher than in untreated fibroblasts. Currently, less is known about matrix composition that might increase matrix stiffening. We previously reported that incorporation of fibronectin into 3D collagen matrices did not change matrix stiffness [11], while the presence of low- and high-molecular-weight HA enhanced the matrix elastic modulus [14,31]. Since activated fibroblastic cells might secrete a broad range of matrix proteins or crosslinking proteins (e.g., lysyl oxidase or transglutaminase) [51] that might contribute to increase of the matrix stiffening as well, we did not further characterize elastic modulus of decellularized matrix in dependence of HA depletion or other influences of composition or crosslinking. We only focused to demonstrate that both cells behave distinctively different in terms of matrix remodeling, and that changes in matrix porosity do not primarily cause changes in matrix stiffness.

Overall, we showed that NHFB and CAF behaved differently in terms of proliferation, aSMA expression, and matrix remodeling. Our data suggests that CAF expressed more aSMA and react more insensitively to TGF-β1 exposure, probably due to pre-stimulation with TGF-β1 in vivo. In addition, CAF appear to have less potential to express matrix proteins and remodel the collagen matrices, when compared to myofibroblast. The characteristics of CAF found in this work are in line with another report [3], describing CAF as low matrix production and remodeling phenotype when compared to wound-healing-related myofibroblasts.

### 2.2. NHFB and CAF Produced Matrix Bound HA in TGF-β1-Dependent Manner

As demonstrated above, both NHFB and CAF behave distinctively different in the 3D collagen matrices in the presence and absence of TGF-β1. TGF-β1 is also known to promote the synthesis of matrix proteins, such as type I and type III collagen, fibronectin, and proteoglycans [3]. However, less is known about the production of HA. As HA is important in many physiological and pathological processes, including wound healing and cancer development, understanding HA production of NHFB and CAF will help us to distinguish their cellular functions and contributions in reshaping of their resided microenvironments.

To study HA production by NHFB and CAF, we first performed quantitative analysis of HA secretion into cell culture supernatant (Figure 3A) and deposition into matrices (Figure 3B) using a commercial HA ELISA kit. The obtained HA amount were normalized to cell number. As shown in Figure 3A, untreated CAF produced significantly higher amounts of HA when compared to NHFB. In the presence of TGF-β1, both cells increase the production of HA and the amount in the cell culture supernatant was within the same range of 180 ng/mL per 10^4^ cells. No significant difference in soluble HA amount could be observed between untreated and treated CAF. In addition to quantify soluble HA amount, we analyzed matrix bound HA from decellularized collagen matrix after matrix digestion. Higher amount of HA in the range of 500–1500 ng/mL/matrix was found, and it was higher than HA in the supernatant (Figure 3B). It could be observed that only TGF-β1 treated CAF produced more matrix bound HA when compared to NHFB and untreated CAF. This data suggests that TGF-β1 can trigger matrix bound HA production which might contribute in an increase of HA in the tumor microenvironments.

To visualize the matrix bound HA, we first decellularized the collagen matrix and stained with hyaluronan binding protein (HABP). Matrices were imaged using confocal laser scanning microscope and representative images of HA bound to collagen matrices are showed in Figure 3C. The optical microscopy confirmed the quantitative analysis that CAF produced a higher amount of matrix bound HA compared to myofibroblasts. However, HA was not homogenously distributed throughout the matrix, but was found as spots, suggesting a local synthesis of HA by cells, as also demonstrated in our previous work [14].

Overall, our results demonstrated that both NHFB and CAF produced a higher amount of matrix bound HA when compared to the soluble counterpart. It has been hypothesized that HA is produced at the adhesion plaques [52] and might bind to the cellular fibronectin that is synthesized simultaneously [53,54]. In addition, TGF-β1 enhances the production of soluble HA in NHFB, while it increases matrix-bound HA in CAF, which appears to be incorporated into collagen fibrils. Our results support a previous finding that human dermal fibroblasts enhance HA production in dependence of TGF-β1 [55], while other reports demonstrated no TGF-β1 dependent increase of HA secretion by dermal fibroblasts [56,57]. It has to be mentioned that low molecular HA (<100 kDa) could not be detected using this ELISA approach, as previously reported [14,58].

### 2.3. Matrix Bound HA Is Medium-Size Molecular Weight

Not only the amount of HA in the microenvironment modulates cellular responses, but also their molecular weight is discussed as an important cellular regulator in many processes including wound healing and cancer progression [14,31]. In addition, the molecular weight of HA can influence the binding of CD44 [31,59] and the downstream signaling [60]. We therefore adapted an agarose gel electrophoresis technique for estimation of the molecular weight of HA in cell culture supernatant and matrix-bound HA. As shown in Figure 4A, we could observe a smear band of HA in the digested matrices, while samples from cell culture supernatant did not show any band. A strong HA stain could be detected in CAF treated with TGF-β1. This is in line with the quantitative analysis (Figure 3A,B) and also points out the low concentration of HA in cell culture supernatant. The observed matrix-bound HA did not show a distinct molecular weight, but a smear band. This can be explained by a broad polydispersity of HA. One has to keep in mind that produced HA can be processed by HA degrading enzymes such as HYAL leading to a variation of molecular weight which in turn is known to regulate physiologically and pathologically relevant cellular functions.

We further determined the average molecular weight of the matrix-bound HA by plotting the band intensity as a function of HA molecular weight. Figure 4B showed an exemplary analysis of the HA molecular weight of CAF treated with TGF-β1. The average molecular weight of all groups is shown in Figure 4C and were approximately 480 kDa for all digested matrix samples. As known, dermal HA is usually found to possess a high molecular weight HA (4000–6000 kDa) [61]. However, the large HA in human skin is rapidly broken down into medium-sized fragments of 300–600 kDa [62]. This is in agreement with our finding of an average molecular weight of HA of approximately 480 kDa in our biomimetic 3D cell culture system.

### 2.4. HAS2 and HYAL1 Are Highly Expressed in CAF

To address which HA metabolic enzymes are involved in the HA synthesis and degradation in our 3D culture models, we analyzed the gene expression of HA metabolic enzymes, namely hyaluronan synthase (HAS) and hyaluronidase (HYAL) using RT-qPCR. As shown in Figure 5A, we found higher expression of *HAS1* and *HAS2* in CAF when compared to NHFB, but not *HAS3* (Figure 5A(iii)). In the presence of TGF-β1, the expression of *HAS1* and *HAS2* were enhanced in NHFB, while only *HAS2* expression was enhanced in CAF (Figure 5A(i) and Figure 5A(ii)). As mentioned, different HAS isotypes produce different HA size, HAS1 and HAS2 produce 200 kDa to 2000 kDa HA, while HAS3 yielded HA product at the average molecular weight of 100 to 1000 kDa [18,19]. Our current results could not determine which HAS isoforms were actively producing HA. However, the found gene expression of *HAS1* and *HAS2* of CAF in this work are in line with the expression reported in many cancer models [14,23,63,64]. Along with our previous work showing that HAS2 knock-out in fibroblasts results in significant reduction of skin HA both in vivo and in vitro models [14], we hypothesize that an increase in HA production by CAF might cause a positive feedback loop of HAS2 in CAF. Loss of HAS1 and HAS3 leads to an enhanced inflammation [65], the expression of HAS1 in NHFB in the presence of TGF-β1 might assist the tissue regeneration process by reduction of pro-inflammatory milieu. In addition, HAS3 is often associated with HA turnover in inflammatory conditions and HAS3-siRNA inhibits neointimal hyperplasia [66]. It has to be mentioned that the expression level might not correlate to the metabolic activity of HAS due to post-translational modifications and nc-RNA mediated regulation [67].

HA can be degraded by major HA catabolic enzymes to a broad spectrum of sizes at a few hundred kDa. Especially, tumors have been proposed to generate very low molecular weight HA [13,14,15]. We analyzed *HYAL1–3* isoforms and found an upregulation in *HYAL1* expression in dependence of TGF-β1 presence in both cell types, however, higher expression could be observed in activated CAF when compared to myofibroblasts (Figure 5B(i)). On the contrary, *HYAL2* and *HYAL3* did not show any change (Figure 5B(ii) and Figure 5B(iii)). The function of HYAL1 in fibroblasts during cancer progression is still unclear. It has been demonstrated that HYAL1 degrades HA to a low molecular weight range that may regulate specific cellular functions [14,31,68,69,70]. It has been reported that average HA fragments (100–400 kDa) produced in vivo are associated with inflammation and cell signaling by HA receptors (e.g., CD44 and RHAMM) that bind small HA [71]. Although HA fragments can originate from hyaluronidases breakdown of high molecular HA, HAS isoenzymes might also be regulated to synthesize specifically smaller HA molecules. It has to be mentioned that we did not further consider other HA catabolic enzymes in this work, for example and transmembrane protein 2 (TMEM2), which also degrades HMW-HA into small fragments (~5 kDa) [72].

## 3. Conclusions

In this work, we demonstrated the distinct behavior of NHFB and CAF in the absence and presence of TGF-β1 in 3D collagen matrices. Although the results suggested a weaker response of CAF towards TGF-β1 in terms of cell proliferation, expression of aSMA and matrix remodeling, higher production of HA in CAF could be observed in response to TGF-β1 when compared to myofibroblasts (TGF-β1 treated NHFB). The average molecular weight of HA was approximately 480 kDa for both TGF-β1 treated NHFB and CAF. From that we finally conclude that CAF have to be considered as a distinctively different fibroblast phenotype than wound healing-associated myofibroblasts, and that HA production and degradation is specifically regulated by CAF, which in turn would place CAF and their HA regulation function central in tumor regulation.

## 4. Materials and Methods

### 4.1. Cell Culture and Differentiation in of 3D Collagen I Matrices

Cancer-associated fibroblasts (CAF) from melanoma biopsies and dermal fibroblasts (NHFB) were isolated from donors with informed consent and after approval of the local ethics committee (171-1402062014) using a tissue dissociating protocol previously described [73]. For CAF, skin adjacent to the tumor (<5 mm) was used, whereas NHFB were prepared from female breast skin. Dispase II (Roche Diagnostics GmbH, Mannheim, Germany) mediated removal of the epidermal sheet was followed by digestion of the dermal compartment with collagenase (Sigma-Aldrich Chemie GmbH, Schnelldorf, Germany). To remove tissue debris the cell suspension was passed through 70 µm filters (BD Biosciences, San Diego, CA, USA). Cells were cultured with DMEM (Biochrom, Berlin, Germany) supplemented with 10% FCS (Biochrom) and 1% ZellShield (Biochrom) at 37 °C, 5% CO2 until confluence. Cells were detached by 0.05% trypsin/0.02% EDTA (Biochrom). For experiments, primary cells between passages 2–4 were used. Collagen matrices were prepared by mixing collagen stock solution (Corning, New York, NY, USA) with 250 mM phosphate buffer (Sigma-Aldrich Chemie GmbH) and 0.1% acetic acid (Sigma-Aldrich) to achieve 2 mg/mL collagen concentration, according to the established protocol [74]. Cells were embedded into 3D collagen matrices by mixing with the prepared collagen solution and pouring onto coated coverslips (13 mm in diameter; VWR, Darmstadt, Germany). In total, 1 × 10^4^ cells were embedded in 50 µL collagen solution. Collagen fibrillation was initiated at standard cell culture condition for 30 min at 37 °C, 5% CO_2_ and 95% humidity. Afterwards, reconstituted 3D collagen matrices were washed 3 times with phosphate buffer saline (PBS; Biochrom) prior culturing in DMEM cell culture media supplemented with 10% FBS and 1% ZellShield (all from Biochrom). For fibroblast differentiation, cells were cultured in DMEM cell culture media supplemented with 10 ng/mL TGF-β1 [11]. Both untreated and TGF-β1 treated cells were cultured for 7 days prior further investigation.

### 4.2. Cell Proliferation

Cell proliferation was determined after 4 days by means of commercial WST-1 assay (Roche, Mannheim, Germany), as previously published [11]. Briefly, cells were rinsed 3 times with Hanks’ balanced salt solution containing calcium and magnesium (Biochrom) and were subsequently incubated for 30 min with 500 μL WST-1 solution (1:10 dilution with cell culture medium) at standard cell culture condition. Supernatants were collected and 100 μL of each supernatant were transferred to 96-well plates (VWR). Absorbance was measured at 450 nm with a multi-well plate reader (Tecan F200; Tecan, Austria). Cell number was evaluated by standard curve. Experiments were performed at least with 4 replicates.

### 4.3. Immunocytochemical Staining and Image Analysis

Cells were rinsed three times with PBS (Biochrom) and fixed in 4% paraformaldehyde (Carl Roth, Karlsruhe, Germany), permeabilized with 0.1% Triton X100 (Roth, Germany) and blocked 2% BSA (all in PBS, Sigma-Aldrich). Afterwards, cells were stained with DAPI (dilution 1:10,000 in PBS; Invitrogen, Darmstadt, Germany) and Phalloidin conjugated with Alexa Fluor 488 (dilution 1:250 in PBS; Invitrogen) and anti-human alpha-smooth muscle actin (clone 1A4) conjugated with eFluor 660 (dilution 1:200 in PBS; eBioscience, Frankfurt, Germany) for 2 h at room temperature. Afterwards, cells were imaged with 40×/NA 1.3 oil immersion objective (Zeiss, Jena, Germany) using confocal laser scanning microscope LSM700 (Zeiss).

For quantitative analysis of αSMA positive cells, at least 50 cells were manually counted for each independent experiment. At least 3 independent experiments with fibroblasts and CAF from 3 different donors were performed for each condition.

### 4.4. Quantitative Analysis of 3D Collagen Matrix Topology and Elasticity

Topological and mechanical parameters of 3D collagen matrices were characterized as previously described [74,75]. For topological analysis, 3D collagen matrices were stained with 50 μM 5-(and-6)-carboxytetramethylrhodamine succinimidyl ester (5(6)-TAMRA-SE) (Invitrogen, Carlsbad, CA, USA) at room temperature for 1 h and rinsed 3 times with PBS (Biochrom). Collagen matrices were imaged using confocal laser scanning microscope LSM700 (Zeiss) using 40×/NA 1.3 oil immersion objective (Zeiss). Pore size was quantified as previously described using a home-built image analysis tool using an erosion algorithm and autocorrelation analysis [75], respectively. The topological analysis was performed at least in triplicates with 4 positions of each Coll I matrix with 3 independent experiments.

Elastic modulus of collagen matrices was determined by colloidal probe force spectroscopy using a scanning force microscope (NanoWizard 3, JPK Instruments, Berlin, Germany), as previously reported [74,76]. In brief, a 50 μm glass microbead (Polyscience Europe GmbH, Eppelheim, Germany) was attached to a tipless HQ-CSC38 cantilever (NanoAndMore, Wetzlar, Germany) with a spring constant of ~0.1 N/m. At least 50 force-distance curves at 3 positions of each collagen with 3 independent experiments was measured in PBS buffer (Biochrom) at room temperature. Young’s modulus of the collagen matrices was determined by fitting the retract part of force distance curves (typical indentation depth 5 μm) using the Hertz model.

### 4.5. Quantitative Analysis of HA Amount in 3D Collagen Matrices

HA amount was quantified using an established protocol [14]. Briefly, collagen matrices were decellularized by incubation with distillated water for 1 h. Afterwards, decellularized collagen matrices were digested with digestion solution consisting of papain solution (0.02 mg/mL papain from papaya latex (Sigma-Aldrich), 10 mM EDTA (Sigma-Aldrich), 5 mM L-cysteine (Sigma-Aldrich) in 5× PBS for 2 h at 60 °C. The amount of HA production was quantified using commercial HA-ELISA kit (TECOmedical Group, Sissach, Switzerland) following manufacturer’s protocol.

### 4.6. Localization of HA in 3D Collagen Matrices

3D collagen matrices were decellularized using distilled water for 1 h at room temperature. HA was visualized by staining with biotinylated hyaluronan binding protein (HABP; final conc. 3 µg/mL; Amsbio LLC, Abingdon, UK) overnight at room temperature. Afterwards, matrices were washed 3 times with PBS (Biochrom). Subsequently, streptavidin conjugated with Alexa Flour 488 (Santa Cruz Biotechnology, Dallas, TX, USA) was used to visualized HA localization within 3D collagen matrices. For the imaging, confocal laser scanning microscopy (Zeiss) with 40x/NA 1.3 water objective (Zeiss) was used.

### 4.7. Gel Electrophoresis

The gel electrophoresis was adapted from Lee et al. [77]. Briefly, 0.5% agarose gel was prepared in Tris-acetate ethylenediaminetetraacetic acid buffer (1× TAE buffer; Sigma-Aldrich). Afterwards, gel was equilibrated in 1× TAE for 15 min. Samples were mixed with a tracking dye composed of 0.02% bromophenol blue in TAE buffer containing 2 M sucrose (Sigma-Aldrich). Samples and HA ladder with specific HA size (Amsbio LLC) were then transfer to the agarose gel. Electrophoresis was carried out at room temperature for 2 h with a constant voltage of 50 V. Subsequently, agarose gel was first incubated for 1 h in 30% ethanol and thereafter in 0.005% Stains-All solution (prepared in 30% ethanol; Sigma-Aldrich) overnight. To remove staining solution, gel was washed with distilled water and incubated in 10% ethanol for 1 h.

### 4.8. Gene Expression Analysis

mRNA isolation and PCR procedures were performed as previously described [11,35]. Briefly, mRNA was isolated with a ReliaPrep™ RNA Tissue Miniprep System (Promega Corporation, Madison, WI, USA) according to manufacturer’s protocol and mRNA amount was determined by UV/VIS spectrophotometry (Eppendorf, Hamburg, Germany). The RNA obtained was converted into complementary DNA (cDNA) using a GoScript™ Reverse Transcriptase kit (Promega, USA). RT-qPCR was performed using GoTaq^®^ qPCR Master Mix (Promega, USA) on a LineGene 9600 (Bioer, Hangzhou China). The primers for *αSMA*, *Coll1a1*, *EDA-FN*, *HAS1*, *HAS2*, *HAS3*, *HYAL1*, *HYAL2*, *HYAL3* and *RPS26* (reference gene) genes were synthesized (Eurofins, Martinsried, Germany) (Table 1). The RT-qPCR procedure was set as follows: hot start for 5 min at 95 °C; 45 cycles of amplification by denaturation (95 °C, 15 s), annealing under primer-specific conditions (20 s), and target-gene specific extension (30 s at 61 °C). Real-time fluorescence measurement was conducted for 20 s at 72 °C after each cycle. The specificity of the PCR products was confirmed by melting curve analysis at the end of the entire run. Genes were normalized to the unregulated reference gene *RPS26*. The relative expression levels were calculated using the 2^−ΔΔCT^ method. Experiments were performed at least in 6 independent experiments with Fb and CAF from 3 different donors each.

### 4.9. Statistical Analysis

Experiments were performed in at least in 3 replicates with fibroblasts and CAF from 3 different donors for each condition. Error bars indicate standard deviation (SD). Levels of statistical significance were determined using one-way ANOVA using Prism 8 (GraphPad, La Jolla, CA, USA). Significance level was set at *p* < 0.05.

## Figures and Tables

**Figure 1 gels-06-00033-f001:**
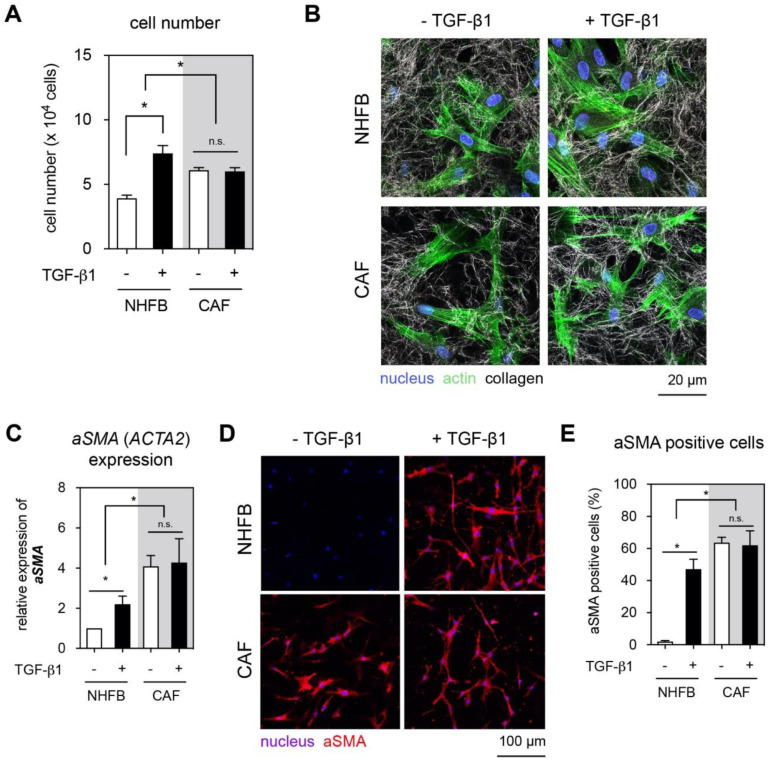
Effect of TGF-β1 on proliferation, aSMA expression, and matrix remodeling of NHFB and CAF. NHFB and CAF were cultured on 3D collagen matrices in the presence and absence of TGF-β1 for 7 days. Both cells were then characterized in terms of proliferation, aSMA expression, and matrix remodeling. (**A**) Quantitative analysis of cell number using commercial WST-1 assay. (**B**) Representative confocal images of NHFB and CAF. Cells were stained with DAPI and Phalloidin conjugated with Alexa Fluor 488 to visualize nucleus and actin, respectively. Collagen fibrils were imaged in the reflection mode. (**C**) The expression of aSMA was analyzed using qPCR and was normalized to NHFB cultured in the absence of TGF-β1. (**D**) Immunocytostaining of aSMA and (**E**) quantitative analysis of number of aSMA positive cells by manual count. Data are represented as mean ± SD; *—significance level of *p* < 0.05; n.s.—not significant. All quantitative experiments were performed at least in 3 replicates with fibroblasts and CAF from 3 different donors for each condition.

**Figure 2 gels-06-00033-f002:**
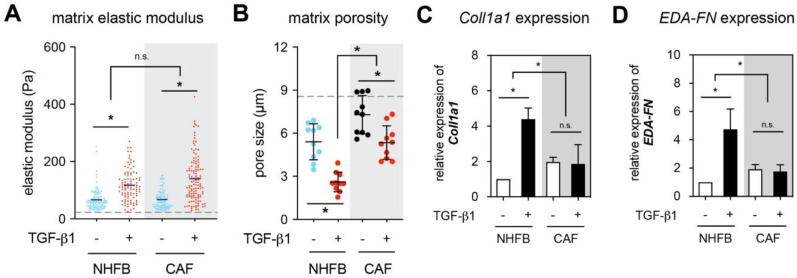
Matrix remodeling of NHFB and CAF in the presence and absence of TGF-β1. Collagen matrices were decellularized and analyzed regarding (**A**) matrix elastic modulus and (**B**) mean pore size. For both (**A**) and (**B**) the dashed line symbolizes empty collagen matrix pore size without cells. The relative gene expression of (**C**) alpha 1 chain of collagen type 1 (*Coll1a1*) and (**D**) fibronectin containing EDA-domain (*EDA-FN*) were analyzed using qPCR and were normalized to NHFB cultured in the absence of TGF-β1. Data are represented as mean ± SD; *—significance level of *p* < 0.05; n.s.—not significant. All quantitative experiments were performed at least in 3 replicates using fibroblasts and CAF from 3 different donors for each condition.

**Figure 3 gels-06-00033-f003:**
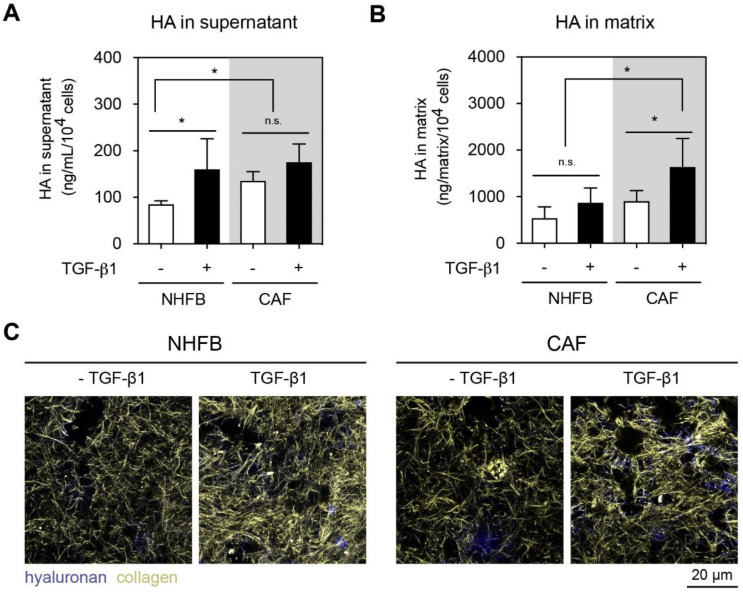
HA production and localization. HA amount of (**A**) cell culture supernatant and (**B**) decellularized matrices were analyzed using a commercial HA ELISA kit. Data are represented as mean ± SD; *—significance level of *p* < 0.05; n.s.—not significant. (**C**) Representative images of decellularized 3D collagen matrices of FB and CAF in dependency of TGF-β1. Matrices were stained with biotin-tagged HABP followed by streptavidin conjugated with Alexa Fluor 488 to visualize HA (dark blue). Collagen fibrils were visualized using reflection mode (grey-yellow). All quantitative experiments were performed at least in 3 replicates using fibroblasts and CAF from 3 different donors for each condition.

**Figure 4 gels-06-00033-f004:**
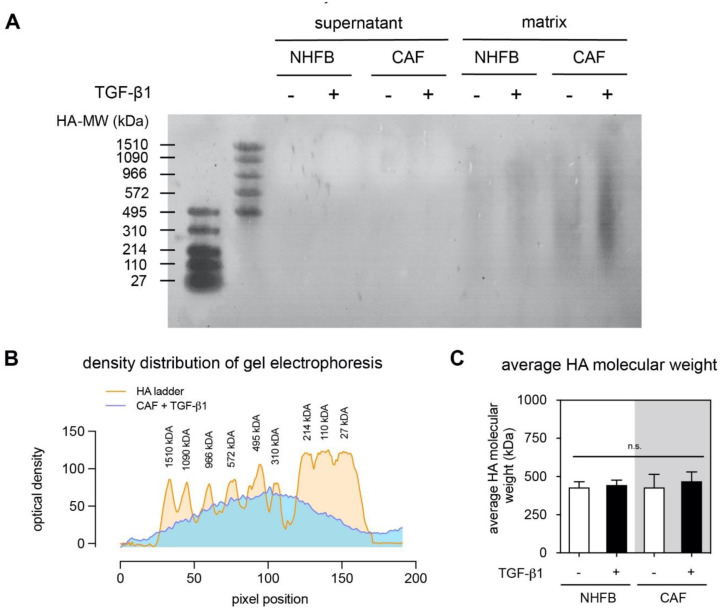
Analysis of HA molecular weight. (**A**) Representative images of electrophoresis gel of supernatant and soluble decellularized collagen matrices of FB and CAF in dependence of TGF-β1. (**B**) Representative density profile plot of molecular weight distribution of matrix bound HA of CAF treated with TGF-β1. (**C**) Analysis of the average HA molecular weight of decellularized matrices. Data are presence as mean ± SD; n.s.—not significant. All quantitative experiments were performed at least in 3 replicates with fibroblasts and CAF from 3 different donors for each condition.

**Figure 5 gels-06-00033-f005:**
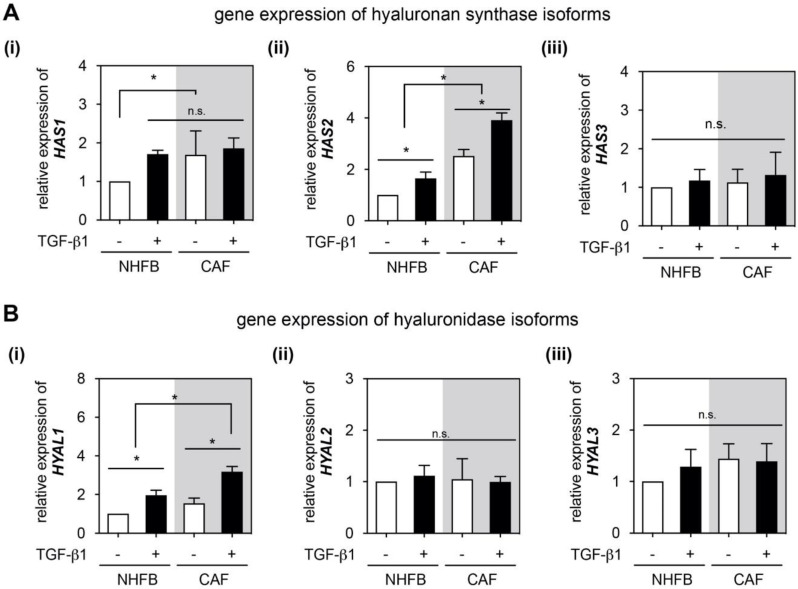
Gene Expression of HA related metabolic enzymes. Quantitative analysis of gene expression of (**A**(**i**)) HAS1, (**A**(**ii**)) HAS2 and (**A**(**iii**)) HAS3, and (**B**(**i**)) HYAL1, (**B**(**ii**)) HYAL2 and (**B**(**iii**)) HYAL3 of FB and CAF in dependence of TGF-β1 presentation. Data are normalized to NHFB without TGF-β1 treatment. Data are represented as mean ± SD; *—significance level of *p* < 0.05; n.s.—not significant. For all quantitative experiments were performed at least 3 replicates with fibroblasts and CAF from 3 different donors for each condition.

**Table 1 gels-06-00033-t001:** RT-qPCR primer sequences used in the experiments.

Primer	Sequence (5′ → 3′)
*RPS26*	
forward	CAATGGTCGTGCCAAAAAG
reverse	TTCACATACAGCTTGGGAAGC
*αSMA (ACTA2)*	
forward	AGACCCTGTTCCAGCCATC
reverse	TGCTAGGGCCGTGATCTC
*Collagen I(α1)*	
forward	GTCGCACTGGTGATGCTG
reverse	GGTGGTGTCCACCTCGAG
*EDA-FN*	
forward	CCAGTCCACAGCTATTCCTG
reverse	ACAACCACGGATGAGCTG
*HAS1*	
forward	GACTCCTGGGTCAGCTTCCTAAG
reverse	AAACTGCTGCAAGAGGTTATTCCT
*HAS2*	
forward	CCAGTCCACAGCTATTCCTG
reverse	ACAACCACGGATGAGCTG
*HAS3*	
forward	GTCGCACTGGTGATGCTG
reverse	GGTGGTGTCCACCTCGAG
*HYAL1*	
forward	CCAGTCCACAGCTATTCCTG
reverse	ACAACCACGGATGAGCTG
*HYAL2*	
forward	GTCGCACTGGTGATGCTG
reverse	GGTGGTGTCCACCTCGAG
*HYAL3*	
forward	CCAGTCCACAGCTATTCCTG
reverse	ACAACCACGGATGAGCTG

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
