# Peer review of "Matrix Remodeling and Hyaluronan Production by Myofibroblasts and Cancer-Associated Fibroblasts in 3D Collagen Matrices"

_gels, 2020, doi:10.3390/gels6040033_

Round 1
Reviewer 1 Report
In this manuscript Sapudom et al have investigated hyaluronan (HA) production and matrix remodeling by (cancer-associated) fibroblasts in 3 dimensional collagen matrices. They further investigated the role of TGFb on cell proliferation, smooth muscle actin (SMA) expression and matrix/HA deposition. Although the subject is interesting, the reported data are not really convincing. Limited data are incorporated into this report, cells are not characterized and many of the differences observed might be cause or by the source of the cells or by differences in the basal SMA expression. This should be carefully addressed. Furthermore, as the authors indicate, many of the reported findings have been observed and published before, thereby limiting the novelty of this study.
Major comments:
- Language editing is required.
- Please indicate the number of independent experiments (biological repeats) in the figure legends. Scatter bar graphs would be helpful to interpret and assess experimental spread of the data.
- In the introduction it is stated that TGFb acts as a tumor suppressor, which is only the case in premalignant stages. In tumor stage cells become insensitive to the growth invasive properties and are stimulated to invade and metastasize by TGFb signaling. Please adjust.
- The authors have isolated CAFs from the normal skin adjacent to the melanoma. This is somewhat surprising. What defines them as CAFs, since they were not derived from the tumor? Furthermore characterization (presence of e.g. vimentin/ %SMA positive cells, FAP, and absence of epithelial, immune and endothelial markers (e.g. cytokeratin/CD45, CD31) of the CAFs and normal fibroblasts is missing. This should be added.
- Figure-1, do these data imply that basal CAF proliferation is higher compared to NHFB? If so, is this dependent on endogenous TGFb production? There are some indication for this looking at the relative SMA expression in figure 1C. An experiment with a TGFb receptor kinase inhibitor could answer this question. Figure 1E not clear what group (control or stimulated) are significant different between NHFB and CAF.
- What is the percentage of SMA positive cells in the respective cultures? Both basal and after TGFb stimulation. This might be different from the relative expression (TGFb might increase the expression in the positive cells, or increase the number of positive cells).
- Line 175; I do not completely agree with this statement. CAFs can, under the influence of TGFb produce many ECM components, including collagen type I, fibronectin and multiple others. The authors do not show sufficient data to support this conclusion.
- Figure 3 is not convincing. Only a smear is seen in the CAFs stimulated with TGFb, where one could doubt if this is specific. The density distribution in figure 3B also does not show a clear pattern of what is going on. Another approach showing specifically HA presence is needed to support their conclusions
- Figure 4; is this a representative experiment with SD representing technical triplicates? The differences reported here are minor, so interpretation of thee data and conclusions based on these data should be carefully drawn.
Reviewer 2 Report
General comments
Authors present an in vitro study where they explore the matrix remodeling and HA production potential of tumor-adjacent fibroblasts and normal skin fibroblasts in the absence or presence of TGF-b1, while embedded in a 3D collagen gel matrices. The study add some new knowledge on the biology of tumor fibroblasts and their role on ECM regulation, however there are specific points that need to be revised by authors before the study can be accepted for publication.
Specific compulsory comments
- About the cells used in the study: Authors have used primary cultures of CAFs and NHFB from surgical specimens. Why authors have used fibroblasts adjacent tumors as CAFs and not intratumoral fibroblasts?. There are numerous studies that have documented differences between intratumoral CAFs and near-tumor CAFs.
- How many donors have been used in this study?. Are the presented results coming from single donor or multiple donors? Are CAFs from different donors behaving in the same way or are there inter-donor differences?. This needs to be specified in results and discussed.
- Authors need to check potential cell activation caused by the 3D collagen culture configuration. Expression of aSMA, HA and collagen needs to be compared betweem 2D cultures and 3D cultures (in the absence of TFGb1) to determine if the 3D collagen induce cell activation. This is especially relevant for the NHFB group.
- Information on the 3D culture generation is missing in M&M: number of cells/well?, volume of collagen/well?, volume of incubation medium/well?, time used for polimerization of the collagen?
- In figure 1B, authors stain the cultures with phalloidin. Since they are interested in checking aSMA expression, why they do not stin the cultuers for aSMA?
- In figure 2C, authors use alexa fluor 488 to stain HA. Why the staining is shown in blue instead of green?
- In the chapter for gene expression analyses (M&M), information is missing. Please check text.
Reviewer 3 Report
The authors investigated the differential behavior of wound healing-associated dermal myofibroblasts and cancer-associated fibroblasts (CAF) in term of proliferation, matrix remodeling potential and hyaluronan (HA) production in 3D cell culture systems. In particular, normal human dermal fibroblasts (NHFB) and CAF were embedded into 3D collagen matrices mimicking the extracellular matrix of a tumor microenvironment and thereafter activated by pro-fibrotic TGF-b1. Cell proliferation, matrix remodeling, HA production and gene expression of HA metabolizing enzymes were analyzed. The results indicate a distinct behavior of NHFB and CAF in 3D collagen matrices in the presence of TGF-b1 in terms of matrix remodeling and HA production suggesting that CAF and their HA regulation may exert a pivotal role in tumor regulation.
Overall, these findings are very interesting.
The reviewer has the following suggestions for the authors:
- The authors investigated cell proliferation by WST-1 assay. Another important characteristic of myofibroblasts is the resistance to apoptosis. It would be very interesting to assess whether wound healing-associated dermal myofibroblasts and CAF may display a different susceptibility to apoptosis.
- In Figure 1B, the authors assessed cell actin cytoskeleton by staining of F-actin with phalloidin. It would be very important to specifically assess the organization of alpha-SMA+ stress fibers in the different experimental conditions by using immunofluorescence.
- In Figure 1C, aSMA should be changed with the gene name (ACTA2).
- Some terminology used should be carefully revised throughout the manuscript. Indeed, the authors often use the term “myofibroblasts” when they refer to NHFB challenged with pro-fibrotic TGF-b1. However, this terminology is confusing, because, as also stated at the beginning of the introduction, even CAF have features of myofibroblasts. For instance, see in the abstract: “However, activated CAF demonstrated higher HA production when compared to myofibroblast”. I suggest to use “wound healing-related myofibroblasts” when referring to NHFB activated with TGF-b1.
- In the title and in different part of the manuscript, "myofibroblast" and "fibroblast" should be changed with "myofibroblasts" and "fibroblasts", respectively.
Round 2
Reviewer 1 Report
The authors have addressed some of my concerns and I understand the technical challenges they are facing. I am happy to see that all experiments have been performed with cells from multiple donors. Below some remaining questions I have:
Comment-4:
If the fibroblasts are not isolated from the tumor, one cannot call the cancer-associated fibroblasts to my opinion. Next to that previous characterisation of the cells, does not mean that new batches of cells pose the same characteristics.
Comment-5:
If the maximum induction of proliferation would already be reached by the basal TGFb production, one would indeed not see an increase by adding exogenous TGFb. Therefore I strongly encourage to perform the experiment.
